# Nitrogen Migration and Transformation Mechanism in the Groundwater Level Fluctuation Zone of Typical Medium

Yuepeng Li [1,2,*], Gang Bai [1,2], Xun Zou [1,2], Jihong Qu [1,2] and Liuyue Wang [1,2]

1     College of Geosciences and Engineering, North China University of Water Resources and Electric Power, Zhengzhou 450046, China; blongtobai@outlook.com (G.B.); zouxun.1996@163.com (X.Z.); qujihong@ncwu.edu.cn (J.Q.); wangliuyue1997@outlook.com (L.W.)
2     Collaborative Innovation Center for Efficient Utilization of Water Resources, Zhengzhou 450046, China
\*     Correspondence: lypncwu@163.com; Tel.: +86-139-3716-5752

**Abstract:** Because of the nitrogen pollution problem in groundwater, the migration conversion mechanism of nitrogen in groundwater level fluctuations was analyzed. Technology and methods through indoor experiments and theoretical analysis were used to study coarse sand, medium sand, and fine sand groundwater level fluctuation in the aeration zone and saturated zone under the situation of nitrogen distribution characteristics, revealing groundwater level fluctuation with the nitrogen migration mechanism. The experimental results showed that the variation range of the nitrate-nitrogen ($NO_3^- - N$) concentration with the water level is medium sand > fine sand > coarse sand. The ammonium nitrogen ($NH_4^+ - N$) concentration showed a downward trend after water level fluctuations, and there were more apparent fluctuations in coarse sand and medium sand. The nitrite nitrogen ($NO_2^- - N$) in coarse sand and medium sand first increased the water level and then gradually reached a balance. The sampling points below the water level in fine sand showed a downward trend with fluctuation of the water level, and then gradually reached equilibrium. The results provide a scientific basis for the remediation and treatment of soil and groundwater pollution.

**Keywords:** groundwater level fluctuation; nitrogen; migration transformation

## 1. Introduction

Due to industrial wastewater, domestic sewage, livestock feces, excessive use of chemical fertilizer [1], atmospheric settlement, and other factors, groundwater nitrogen pollution is severe; it has become a global environmental issue [2–4]. According to the 2018 Bulletin of China's ecological environment, among the 10,168 national groundwater quality monitoring points in China, class IV water quality monitoring points account for 70.7% and class V accounts for 15.5%. Class I~class III accounts for only 13.8%, in which nitrogen (nitrite nitrogen, nitrate nitrogen, and ammonia nitrogen) is the primary pollutant. The research on nitrogen pollution has attracted extensive attention from around the world [5]. The migration and transformation of nitrogen in the aeration zone and saturated zone has also become a hot issue in environmental science, soil science, and groundwater science at home and abroad [6].

Affected by natural conditions and human activities, the groundwater level has significant dynamic variation characteristics. The changes in meteorological and hydrological conditions and human activities, such as groundwater exploitation, can cause the groundwater level, and there is a groundwater level wave band [7]. Under groundwater level fluctuations, the aeration zone–saturated zone interface and capillary strips continue to change so that environmental factors, such as water power conditions, physical chemistry, and biological effects, vary with water levels, which in turn affect the migration of pollutants.

In the 1990s, people paid attention to this aspect of groundwater level fluctuation [8]. Affected by water level fluctuations, the activity of diesel-degrading bacteria in soil is

enhanced, conducive to reducing the diesel content in soil [9]. The research results of Chris C. Tanner et al. showed that water level fluctuations are conducive to the removal of TN and $NH_4^+-N$ in an artificial wetland ecosystem, and the removal efficiency is significantly related to the fluctuation frequency [10]. At the beginning of the 20th century, Cari K et al. studied the removal ability of nitrate in a wetland system [11]. Massei Kamon et al. reviewed the migration characteristics of LNAPL in porous media and carried out LNAPL migration experiments and numerical studies under water level fluctuations [12]. The degree of the complex hysteresis effect of the S-P curve during the alternate drying and moisturization of the water level is reduced due to the initial water saturation of the moisture process in the cycle [13]. In recent years, research on the migration of pollutants in groundwater level fluctuations has concentrated in the migration of NAPL and heavy metals in porous medium. For example, through the establishment of an indoor soil column experiment to study the migration law of benzene series pollutants in soil by water level fluctuation, three different combinations of soil layer physical models were established to study the impact of water level fluctuation on the redistribution of crude oil in the heterogeneous soil layer [14,15]. Through the long-term monitoring of monitoring wells and rivers, it is concluded that groundwater level fluctuation will affect the concentration of arsenic in groundwater in Hanjiang Plain [16]. A sand column experiment was established to study the effect of water level fluctuation on the migration and transformation of iron [17]. Relatively little research on nitrogen in water level fluctuations has been conducted. Water level fluctuations are obtained by establishing water levels and fluctuating two indoor earth column experiments. Water level fluctuations are advantageous for nitrogen migration, which will increase nitrate pollution in groundwater [18]. Liu Jing et al. found that water level fluctuations and nitrogen application jointly affect the content of water level fluctuations, and the concentration of nitrate-nitrogen increases with the increase of nitrogen, and the water level fluctuations promote downward migration of nitrate-nitrogen [19]. Liu Xin et al., through the establishment of an indoor sand tank experiment, found that the increase of shallow groundwater leaches pollutants in the aeration zone, leading to an increase of the nitrate-nitrogen and ammonium nitrogen content in groundwater, and the smaller the buried depth of groundwater, the higher the content of nitrate-nitrogen and ammonium nitrogen [20].

Domestic and foreign scholars have related to the mechanism of nitrogen migration conversion [21,22], but the research on the mechanism of nitrogen migration and transformation under groundwater level fluctuation is still not systematic, with a single soil medium in research objects, which cannot reflect the differences regarding nitrogen migration and transformation between different media under the condition of water level fluctuation, and the pollutants mainly considered are nitrate nitrogen.

In this paper, this study selected three typical media of coarse sand, medium sand, and fine sand to set up soil column experiments with three different media to analyze the migration and transformation mechanism of nitrate, ammonium, and nitrite nitrogen under the influence of water level fluctuation in these three media. The provided data support the simulation and prediction of groundwater level fluctuations on the migration and transformation of three nitrogens, and provide theoretical support for groundwater nitrogen pollution prevention and control.

## 2. Materials and Methods

### 2.1. Experimental Materials

In this experiment, the soil medium was collected on the Bank of the Yellow River in Mengjin District, Luoyang City, Henan Province. There is no plant cover on the riverbank, eliminating the influence of plant activities. It was replenished by the river all year round, and the groundwater level fluctuated obviously. Considering the typicality of the medium and the convenience of sampling, the soil was selected as the experimental soil. The collected soil was then placed on a clean dry geotextile to dry naturally. When the soil was fully air-dried, all the soil was first screened with a 1 mm diameter sieve to filter out tree

roots, grassroots, and other sundries, and then fine sand (0.10 mm < diameter < 0.25 mm), medium sand (0.25 mm < diameter < 0.5 mm), and coarse sand (diameter > 0.5 mm) were screened as experimental soil with sieves with diameters of 0.1, 0.25, and 0.5 mm, respectively. The practical water used was laboratory ultrapure water mechanism water. The composition of the soil particles and their parameters are shown in Table 1.

**Table 1.** Soil particle composition and parameters.

| Sample | Particle Size/mm | PH | Organic Matter Content/(g·kg$^{-1}$) |
|---|---|---|---|
| coarse sand | 0.5–1.0 | 8.5 | 0.241 |
| medium sand | 0.25–0.5 | 8.8 | 0.587 |
| fine sand | 0.125–0.25 | 9.3 | 1.070 |

*2.2. Experimental Device*

The experiment was designed with the same organic glass column, with an inner diameter of 20 cm, a height of 110 cm, and an opening at the top. The bottom was respectively connected with a pressure-measuring tube and a Markov bottle through a three-way valve. A peristaltic pump was installed between the Markov bottle and the column to control the water level to simulate the process of groundwater fluctuation (rise-drop). The pressure-measuring tube was fixed on the soil column, and the organic glass column had four sampling points from the top to the bottom, and the distance from the bottom of the column was 15, 25, 35, and 45 cm, and an adjacent point interval of 10 cm. The sampling points had a Rhizon Soil Solution Sampler (inner diameter: 1 mm, Manufacturer: Shanghai Saifu Biological Technology Co., LTD. Shanghai, China). The top and bottom of each column were paved with 5 cm quartz sand (diameter 2~3 mm) to ensure constant rise and fall of the water level. The experimental device is shown in Figure 1.

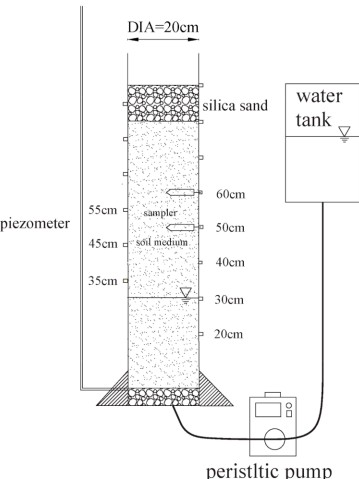

**Figure 1.** Schematic diagram of the experimental device.

*2.3. Experimental Design*

The soil column experiment device is shown in Figure 2. Within the three pillars, the same quality of soil for the different mediums was used, each filled 10 cm, repeatedly compacted with a certain degree of strength, packing the soil medium to the scale, and then filled with the subsequent filling, and a total of about 70 cm soil medium. The soil solution sampler was placed at 15, 25, 35, and 45 cm. The column mouth upper was bound with plastic wrap to prevent moisture evaporating. Distilled water was injected into the column from the marsh bottle through a peristaltic pump to stabilize the water level at a 30 cm scale. The initial water level stood for one week to meet the capillary effects of the earth column and the requirements of the new environment. There was a small pipe connected to the air at the mouth of the marsh bottle. The pressure measuring tube on the side of the

soil column was observed to control the water. The 1 L configuration had a concentration of 500 mg/L of potassium nitrate solution and a concentration of 500 mg/L and 1 L of ammonium chloride solution, and the solution had quartz sand homogeneously dumped on the top of the soil column and an injection water level rise after using the peristaltic pump pumping water with the soil column stable internal water at 30 cm for three days. After three days in each sampling point of the water samples, determination of three initial nitrogen concentrations of each sampling point was conducted. After the measurement, the water level changed periodically, and the time to stabilize the water level was 3 days. Twenty-four days was a complete periodic change, and the experiment was carried out for two periods, as shown in Figure 3 below. The flow of inlet and outlet water was controlled at 100–200 mL·min$^{-1}$. Each time the water level changed, the soil solution sampler was connected by a diaphragm vacuum pump (Model: GM-0.5B; Manufacturer: Tianjin Jinteng Experimental Equipment Co., Ltd., Tianjin, China) or a syringe, and the water sample was extracted at the four sampling points, measuring the pH, temperature, dissolved oxygen, ammonium nitrogen, nitrate nitrogen, and nitrite nitrogen concentrations at each sampling point.

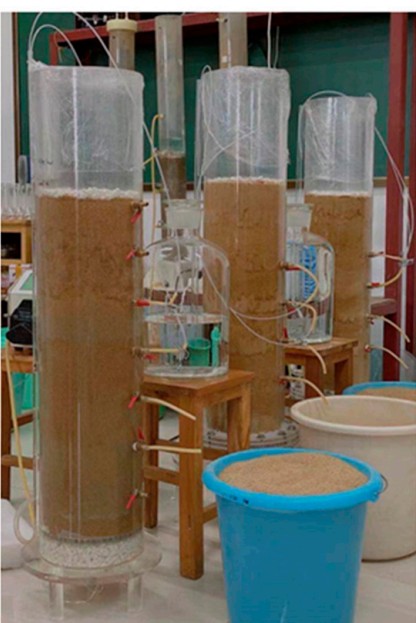

**Figure 2.** Experimental device.

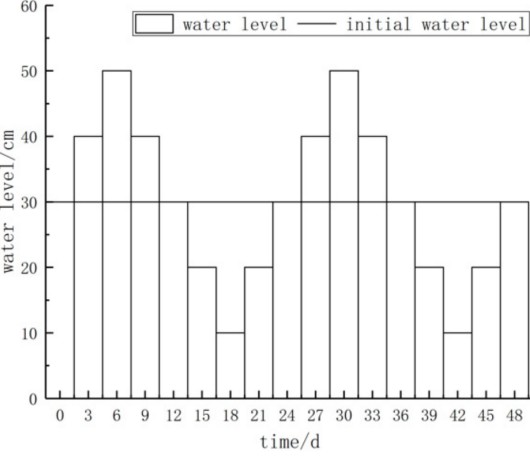

**Figure 3.** Time-variation of the water level.

*2.4. Experimental Method*

$NO_3^- - N$, $NO_2^- - N$, and $NH_4^+ - N$ were detected by Nessler reagent spectrophotometry, ultraviolet spectrophotometry, and N-(1-naphthyl)-ethylenediamine spectrophotometry, respectively, and measured by an ultraviolet spectrophotometer (Model: UV-2550; Manufacturer: Shimadzu (Shanghai) Global Laboratory Consumables Co., Ltd., Shanghai, China). The pH value and temperature were measured by a pH meter (Model: Micro600; Manufacturer: Palintest Co., Ltd., Beijing, China) and thermometer. Dissolved oxygen was measured by a dissolved oxygen meter (Model: JPBJ-608; Manufacturer: Shanghai Yifen Scientific Instrument Co., Ltd., Shanghai, China).

## 3. Results

*3.1. Variation of pH and Dissolved Oxygen in the Water Level Fluctuation Zone*

Temperature, pH, and dissolved oxygen are three essential indicators for studying soil microbial nitrification [23]. This experiment was carried out in the room, and the temperature of each soil column fluctuated in the range of $18 \pm 3$ °C. The pH fluctuated between 7.5 and 8.2, which is generally weakly alkaline, and the dissolved oxygen content is negatively correlated with water level fluctuation. The dissolved oxygen content and pH values in each medium soil solution are shown in Figure 4. In the coarse sand medium, in the rising stage of the water level at sampling point 1, the declined ranges of dissolved oxygen are 38.69%, 54.36%, and 38.41%, respectively. In the falling phase of the water level, the rising fields of dissolved oxygen are 95.73% and 97.27%, respectively. The decreased ranges of dissolved oxygen in sampling point 2 are 43.97%, 41.41%, and 35.90%, and the increased ranges are 102.73% and 65.56%, respectively. The decreased ranges of dissolved oxygen at sampling point 3 are 46.69%, 28.06%, and 1.80%, respectively, and the increased ranges are 58.81% and 25.85%, respectively. In the sand medium, the decreased ranges of dissolved oxygen at sampling point 1 are 49.20%, 40.99%, and 28.83%, respectively, and the increased ranges are 51.41% and 37.54%, respectively. The decreasing fields of sampling point 2 are 37.61%, 47.94%, and 44.17%, and the increasing ranges are 66.18% and 94.33%, respectively. Sampling point 3 decreases by 32.59%, 23.55%, and 33.23%, and increases by 14.86% and 59.60%, respectively. In the fine sand medium, the declined ranges of sampling point 1 are 36.78%, 32.51%, and 27.47%, respectively, and the increased ranges are 35.15% and 37.87%, respectively. The decreasing ranges of sampling point 2 are 36.79%, 35.70%, and 13.54%, respectively, and the increasing ranges are 43.22% and 35.89%, respectively. The decreasing ranges of sampling point 3 are 33.44%, 27.88%, and 8.68%, respectively, and the increasing ranges are 31.75% and 20.70%. The declined ranges of sampling point 4 are 27.22%, 28.43%, and 21.51%, respectively, and the increased ranges are 16.34% and 43.97%, respectively. The dissolved oxygen content in coarse sand fluctuates most violently with the water level, and the reaction of medium sand is more vital than that of fine sand. This is because the particle size and porosity of coarse sand and medium sand are large, and the oxygen content in the aeration zone is more than that in fine sand. The increase of the groundwater level carries the pore water in the saturated zone upwards, thereby intercepting the air in the unsaturated zone, and dissolving into the groundwater under the action of hydrostatic pressure, increasing the dissolved oxygen in the groundwater [24]. As the sampling depth increases, the further the sampling point is from the unsaturated zone, the less oxygen dissolved. Among the three media, the pH of fine sand fluctuates most violently with the water level. When the water level rises from 18 to 24 days, the pH increases significantly but decreases significantly when the water level increases from 24 to 27 days.

The pH change of each sampling point is shown in Figure 5. With the increase of the sampling height, the pH value of coarse sand and medium sand increases, while fine sand decreases. The average pH value of coarse sand and medium sand in the first cycle is more significant than that in the second cycle.

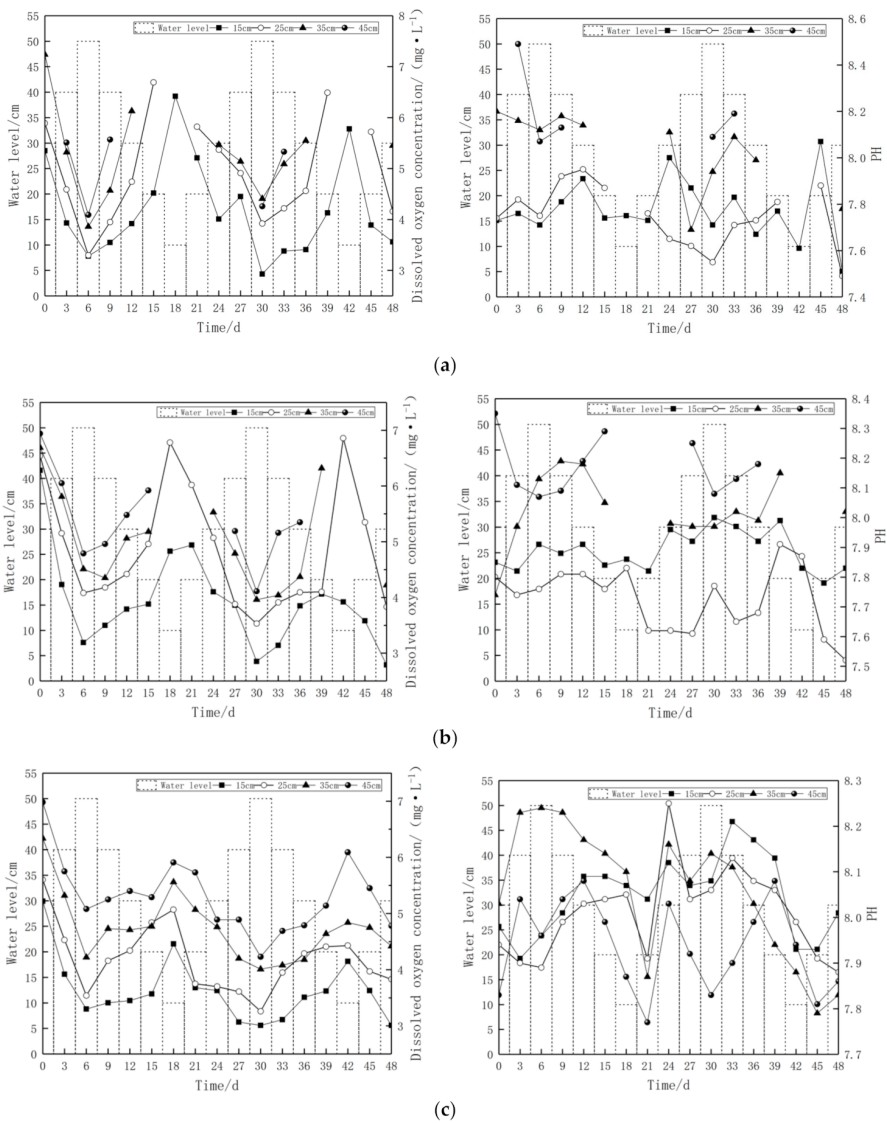

**Figure 4.** Changes in pH and dissolved oxygen in different medium ((**a**) coarse sand, (**b**) medium sand, (**c**) fine sand).

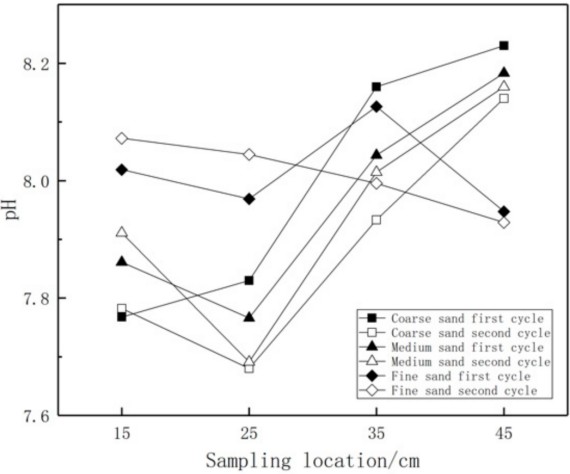

**Figure 5.** Average pH value at various locations on soil columns with different media.

*3.2. Analysis of Nitrogen Migration Conversion*

3.2.1. Nitrate Nitrogen Change Law

Under water level fluctuations, different soil media, and different positions, the $NO_3^- - N$ change process is shown in Figure 6. It can be seen from the change process curve of $NO_3^- - N$ that the concentration of $NO_3^- - N$ at sampling points 1 and 2 in the fine sand medium is only 0.2~2.5 mg/L, which is far less than the concentration of $NO_3^- - N$ at the same sampling points in coarse sand and medium sand medium. $NO_3^- - N$ is mainly distributed at sampling points 3 and 4. The variation of the $NO_3^- - N$ concentration in fine sand and medium sand at sampling points 1 and 2 is more significant than that of coarse sand with the fluctuation of the water level. The variation of the $NO_3^- - N$ concentration in fine sand at sampling point 3 is more significant than that of medium sand. The interpretation of the $NO_3^- - N$ concentration in fine sand at sampling point 4 is unchanged with the water level fluctuation. In a coarse sand medium, the concentration of $NO_3^- - N$ at sampling points 1 and 2 shows an overall downward trend, and shows a downward trend from 0 to 6 days, with a declining range of 8.16% and 5.20%. It shows an upward trend from 6 to 18 days, with an increasing range of 4.62% and 5.27%; It offers a downward trend from 18 to 30 days, with a decreased range of 21.70% and 22.86%. It shows an upward trend from 30 to 42 days, with an increasing range of 11.02% and 15.47%. It offers a downward trend from 42 to 48 days, with a decreased range of 12.57% and 14.94%. In a medium sand medium, the concentration of $NO_3^- - N$ at sampling points 1 and 2 fluctuates obviously with the water level. The concentration of $NO_3^- - N$ in sampling points 1 and 2 shows a downward trend from 0 to 6 days, with a decreased range of 84.43% and 76.87%. It shows an upward trend from 6 to 18 days, with an increasing range of 925.11% and 788.08%. It offers a downward trend from 18 to 30 days, with a declining range of 82.91% and 50.2%, respectively. The concentration of $NO_3^- - N$ at sampling point 1 shows an upward trend from 30 to 42 days, with an increasing range of 635.48%. It offers a downward trend from 42 to 45 days, with a decreased range of 30.47%. Sampling point 2 shows an upward trend from 30 to 48 days, with an upward range of 92.49%. The concentration of $NO_3^- - N$ at sampling points 1 and 2 shows an upward trend as a whole. In the fine sand medium, the concentration of $NO_3^- - N$ at sampling points 1, 2, and 3 obviously fluctuates with the water level, and the concentration of $NO_3^- - N$ at sampling point 3 changes more violently. During 0~6 days of the water level rise at sampling points 2 and 3, the concentration of $NO_3^- - N$ decreases by 72.73% and 20.38%, respectively. In 6~18 days, the concentration of $NO_3^- - N$ increases by 240% and 627.99%, respectively. From 18 to 30 days, the concentration of $NO_3^- - N$ decreases by 87.5% and 56.74%, respectively. Sampling point 2 shows an upward trend of 354.9% in 30~43 days and a downward trend of 23.7% in 43~48 days. Sampling point 3 shows an upward trend in 30~48 days, with an increasing range of 148.83%. The concentration of $NO_3^- - N$ at sampling point 1 reaches the maximum value of 1.18 mg/L at 15 days and then shows a downward trend. The sampling points below the water level of the three media are affected by the water level fluctuation, while the $NO_3^- - N$ concentration at the sampling points above the water level has no significant change characteristics and is less affected by the water level fluctuation. In the rising stage of the water level, the $NO_3^- - N$ concentration shows an apparent downward trend. In the falling phase of the water level, the $NO_3^- - N$ concentration at the sampling points below the water level shows an apparent upward trend. The variation of the $NO_3^- - N$ concentration with the water level is medium sand > fine sand > coarse sand. This may be due to the decrease of the dissolved oxygen content in the rising stage of water level fluctuation. The reduction of the water level led to the increase of the soil dissolved oxygen concentration in the profile. Therefore, combined with the pH value and dissolved oxygen measurement results, as the water level rises and the duration is prolonged, the dissolved oxygen and pH indicators in the soil change, leading to an increase in the abundance of anaerobic microorganisms. The abundance of denitrification functional genes gradually increases, the nitrification reaction is affected, and the $NO_3^- - N$ content of the soil solution will decrease, resulting in the accumulation

of ammonium nitrogen [25]. As the water level drops and the duration is extended, the abundance of aerobic microorganisms increases. The abundance of nitrification functional genes increases, nitrification increases, and the soil $NO_3^- - N$ concentration increases.

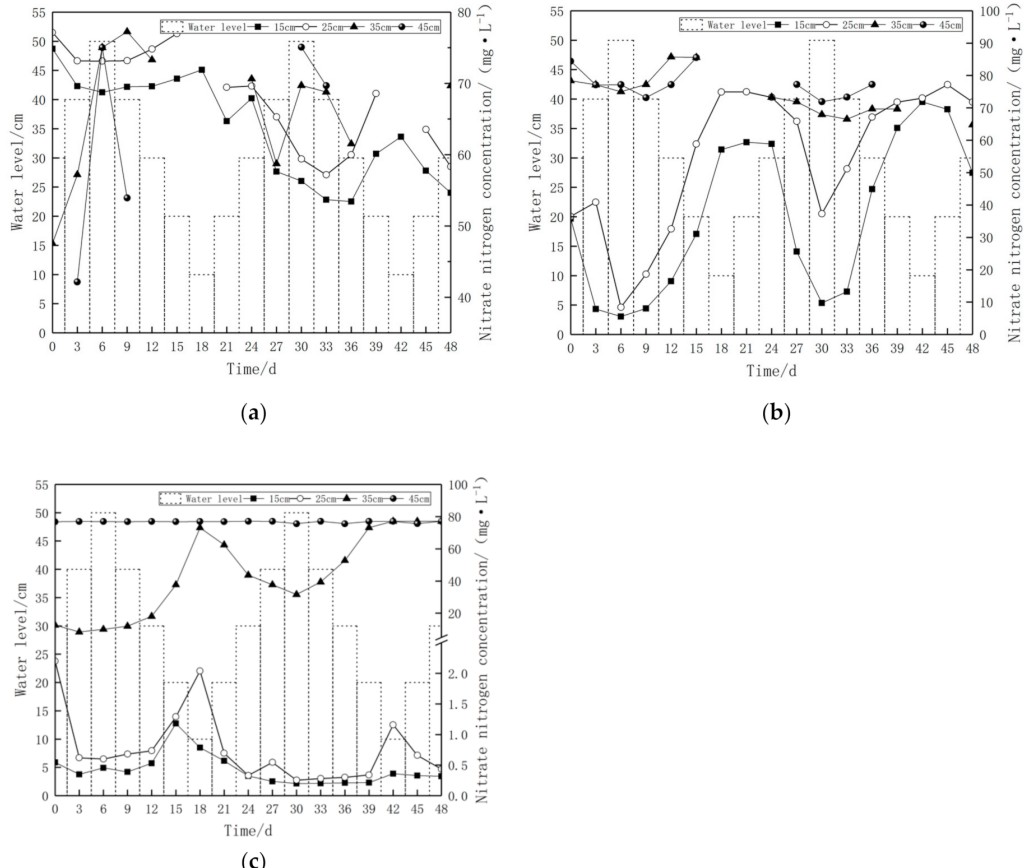

**Figure 6.** Change of nitrate nitrogen in different media ((**a**) coarse sand, (**b**) medium sand, (**c**) fine sand).

When $NO_3^- - N$ moves with water flow in soil, the soil physical and chemical properties, such as the soil water content, water flow movement state, dissolved oxygen content, and microbial community, will affect its migration and transformation process, including adsorption, denitrification, and other reactions [26]. In this study, the soil water content, water flow state, and dissolved oxygen content in the profile below the water level fluctuate sharply with the water level, which is also the main reason for the significant change of the $NO_3^- - N$ content in the soil solution in this profile. The area above the water level is different. Due to the weak adsorption capacity of coarse sand soil to $NO_3^- - N$, $NO_3^- - N$ moves downward rapidly. In addition, the denitrification above the water level is vital, so the $NO_3^- - N$ concentration above the water level is small. The water level fluctuates upward, and a large amount of $NO_3^- - N$ below the water level moves upward. At this time, the physical effect caused by the rise of the water level is more significant than the denitrification, so $NO_3^- - N$ tends to increase. On the contrary, fine sand has a strong adsorption capacity for $NO_3^- - N$ and $NO_3^- - N$ above the water level, which basically does not migrate downward and has a weak response to groundwater level fluctuation.

### 3.2.2. Ammonium Nitrogen Change Law

The change process of $NH_4^+ - N$ is shown in Figure 7. The content of $NH_4^+ - N$ in the three media shows an overall downward trend. The concentration change of $NH_4^+ - N$ at sampling points 1 and 2 is more gentle than that at sampling points 3 and 4. The concentration of $NH_4^+ - N$ reaches a peak at each sampling point after 6 days. In the fine sand medium, the content of $NH_4^+ - N$ in sampling points 1 and 2 ranges from

0.02 to 0.13 mg/L, and $NH_4^+-N$ mainly concentrates in sampling points 3 and 4. The adsorption and desorption experiments show that coarse sand, medium sand, and fine sand reach the adsorption equilibrium within 60 min, and the adsorption capacity of fine sand is the largest. Therefore, the $NH_4^+-N$ in the fine sand medium cannot easily to migrate downward and is mainly concentrated in the upper layer of the soil. In the coarse sand medium, the concentration of $NH_4^+-N$ in sampling points 3 and 4 is less than the concentration of $NH_4^+-N$ in the same position in the medium sand. From 0 to 6 days, sampling points 1 and 2 show an upward trend, with an increase of 29.47% and 18.45%. It shows a downward trend from 6 to 18 days, with a decrease of 22.26% and 13.96%. From 18 to 30 days, a rising trend, with an increase of 1.10% and 16.00%, is shown. A downward trend is shown from 30 to 48 days, with a decrease of 7.20% and 13.90%. The change of the $NH_4^+-N$ concentration in medium sand and fine sand is gentler than that of coarse sand. The range of the concentration change in the rising stage of the water level in the first cycle is significantly greater than that in the second cycle. In the medium sand medium, the first cycle of each sampling point increases by 45.04%, 65.67%, 7.41%, and 9.44%, and the decreased range is 62.34%, 59.04%, 11.22%, and 12.54%. From 30 to 42 days, the sampling points show a downward trend, with the declined ranges of 37.60%, 42.70%, 11.20%, and 16.40%. From 42 to 48 days, sampling points 1 and 2 show an upward trend, with an upward range of 22.98% and 21.42%. In the fine sand medium, the concentration of $NH_4^+-N$ is not significantly affected by the fluctuation of the water in the second cycle. Sampling points 1, 2, and 3 reach peak values of 0.11, 0.13, and 7.57 mg/L on the 6th day, and then show a downward trend from 6 to 48 days, with a declining range of 73.50%, 74.80%, and 67.90%. Sampling point 4 shows a downward trend as a whole, with a decrease of 23.6%, and is slightly affected by water level fluctuations. This may be because the sampling point was located in the vadose zone for a long time with sufficient dissolved oxygen, and nitrification was dominant. During the whole experiment, the fluctuation of the water level has a particular effect on $NH_4^+-N$. When the water level rises, the $NH_4^+-N$ concentration at each sampling point increases. When the water level drops, the $NH_4^+-N$ concentration decreases. The $NH_4^+-N$ concentration and the water level basically fluctuate in the same trend. In the three media, the $NH_4^+-N$ concentration is affected by the water level fluctuation. The amount of change is coarse sand > medium sand > fine sand.

　　Combined with the measurement results of the pH value and dissolved oxygen, it can be concluded that when the water level rises, the dissolved oxygen concentration decreases and the $NH_4^+-N$ concentration of soil solution increases. The water level decreases, the dissolved oxygen concentration increases, and the $NH_4^+-N$ concentration in soil decreases. This is mainly due to the rise of the water level, and the dissolved oxygen content decreased from 6–7 $mg \cdot L^{-1}$ to 3–4 $mg \cdot L^{-1}$. The transformation of the profile from the aerobic stage to the anoxic stage, the gradual activity of denitrifying bacteria, and the gradual enhancement of reducibility resulted in the increase of the $NH_4^+-N$ concentration and the decrease of the $NO_3^--N$ concentration. On the other hand, the rise of the water level will promote the desorption and leaching of $NH_4^+-N$ in soil and make it fully dissolve in water. When the water level drops, the capillary effect in the unsaturated zone weakens, the content of free water in soil pores decreases, the saturation decreases, and the contact area between pore water and air increases, resulting in the growth of the dissolved oxygen content, enhancement of oxidation, promotion of nitrification, and the nitrification being stronger than denitrification, reducing the concentration of $NH_4^+-N$. In addition, the soil has strong adsorption on $NH_4^+-N$, which is not conducive to the accumulation of $NH_4^+-N$ in soil-free water [27], so the $NH_4^+-N$ concentration shows a downward trend as a whole. This is consistent with the research results of Li Xiang et al. on the changes of $NH_4^+-N$ and $NO_3^--N$ concentrations with the fluctuation of the water level [28]. They believe that the rise of the water level leads to a decrease of the dissolved oxygen content, the profile changes from the aerobic stage to the anoxic stage, denitrification leads to a reduction of the $NO_3^--N$ concentration, the accumulation of $NH_4^+-N$, and a reduction of the water

level leads to an increase of the soil dissolved oxygen concentration and enhancement of nitrification. The concentration of $NO_3^- - N$ in soil increased, and the removal of $NH_4^+ - N$ increased.

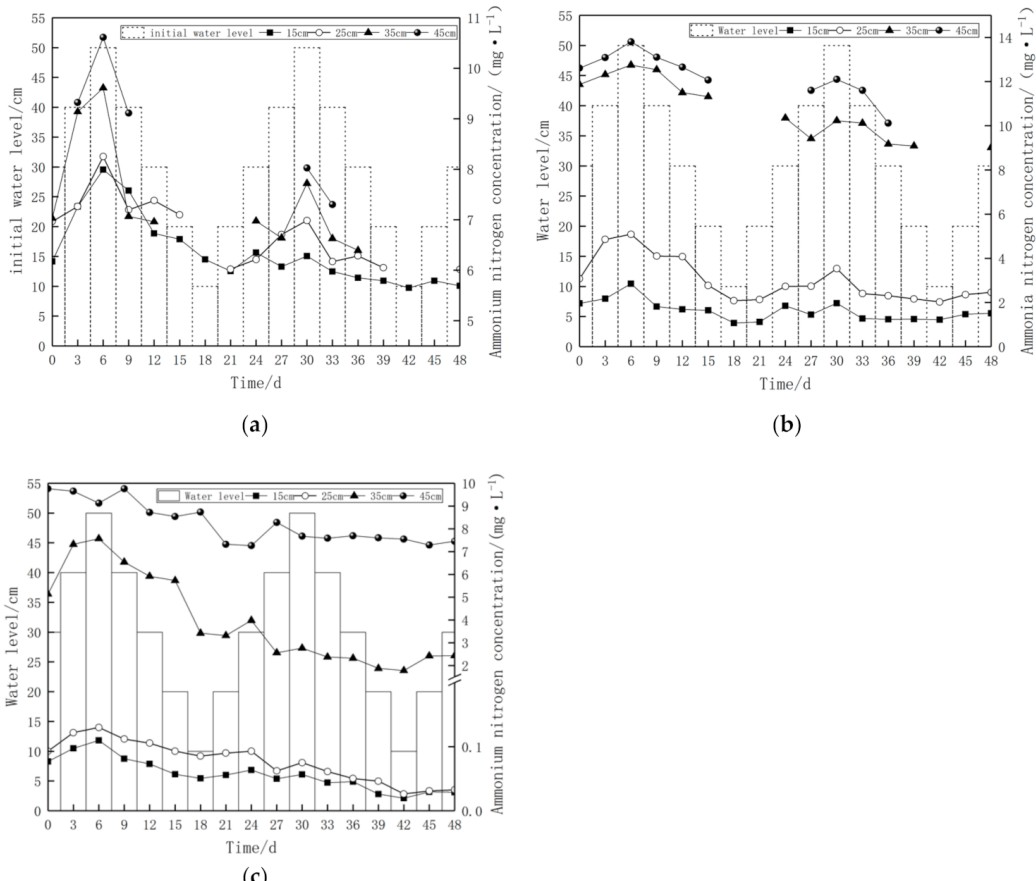

**Figure 7.** Change of ammonium nitrogen in different media ((**a**) coarse sand, (**b**) medium sand, (**c**) fine sand).

### 3.2.3. Nitrite Nitrogen Change Law

Nitrification and nitrosation will produce an intermediate product $NO_2^- - N$, and the concentration of $NO_2^- - N$ is far less than that of $NO_3^- - N$ and $NH_4^+ - N$. In Figure 8, after the significant increases in the first cycle, the $NO_2^- - N$ concentration in coarse sand and medium sand fluctuates at 0.5 mg/L. The $NO_2^- - N$ concentration at sampling points 1 and 2 in fine sand shows a downward trend as a whole and finally, fluctuates at 0.5 mg/l. In the stage of 0–6 days of the water level rise, the content of $NO_2^- - N$ in coarse sand and medium sand increases significantly, which may be because before the water level rise, under the condition of sufficient dissolved oxygen, the growth rate of nitrifying bacteria is much lower than that of nitrobacteria, which makes the nitrosation reaction rate better than the nitrification reaction rate, and finally leads to the increase of the $NO_2^- - N$ concentration [29]. After the water level rise, when the dissolved oxygen content decreases, the nitrification reaction will be affected. In the stage of the water level falling, the concentration of dissolved oxygen in the profile soil increases, the nitrification is enhanced, and the nitrification reaction will produce H+, resulting in the weak acidity of the solution [30]. It is also pointed out that the pH value has a significant negative correlation with the nitrate content but has no significant correlation with ammonium salt and nitrite [31]. Specifically, the solution is weakly acidic, which will enhance the conversion of nitrite to nitrate. It has no significant effect on ammonium salt and nitrite, resulting in an accumulation of the nitrite content and a gradual decreasing. Therefore, the content of $NO_2^- - N$ in the experiment finally tends to fluctuate and bal-

ance. It is evident from Figure 8 that the increase of $NO_2^- -N$ in the initial stage is coarse sand > medium sand > fine sand because the larger the particle size, the more the microbial flocs can contact the particle surface, and the more microorganisms involved in the nitrification reaction, the stronger the nitrification reaction. Therefore, it can be understood that the nitrification ability of microbial flocs will increase with the increase of the particle size [32].

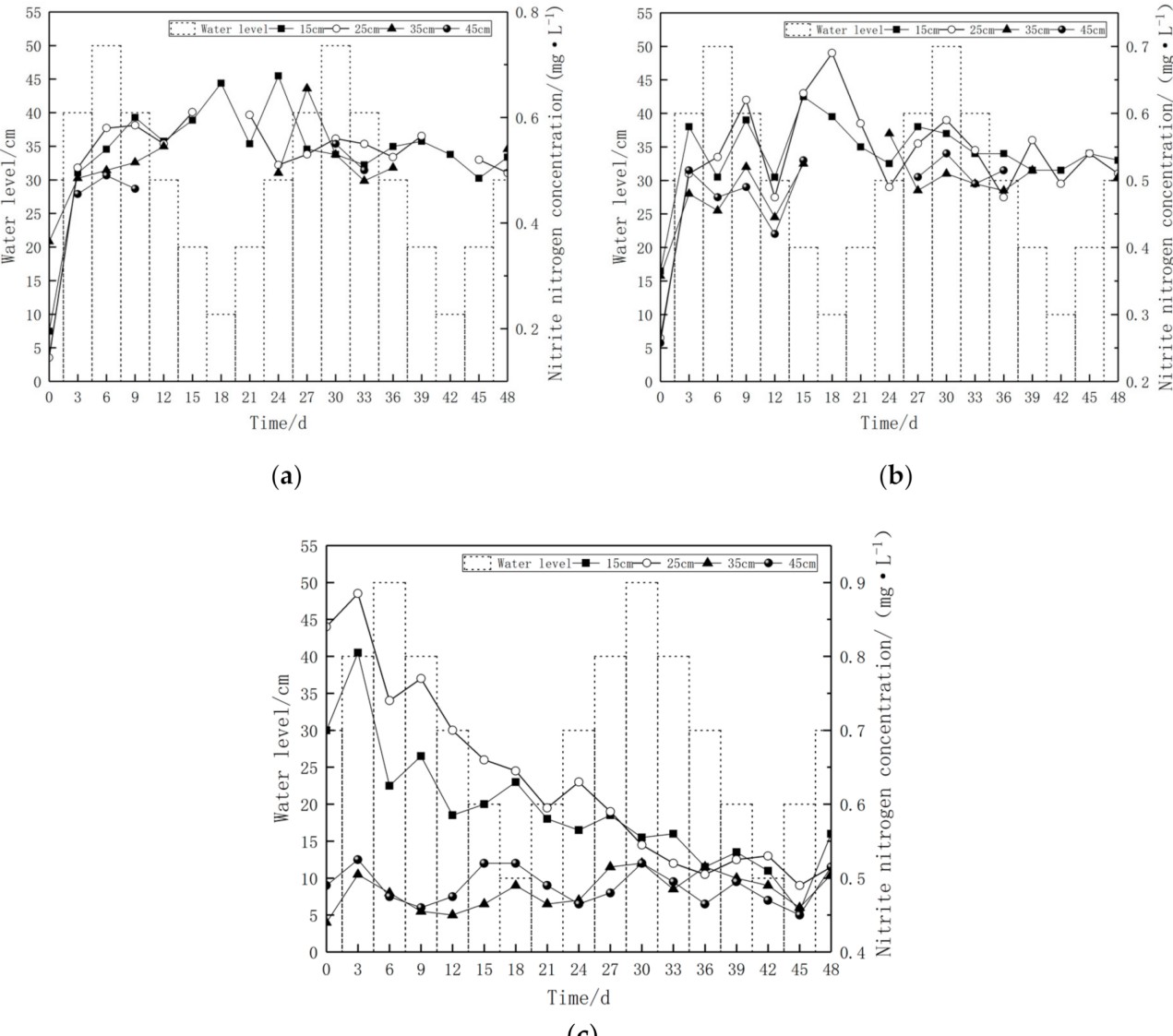

**Figure 8.** Change of nitrite nitrogen in different media ((**a**) coarse sand, (**b**) medium sand, (**c**) fine sand).

## 4. Discussion

Groundwater levels rise and fall in response to climate and human activity, and as they rise and fall, the critical environmental elements in the soil change. In the process of water level change, the groundwater exchange rate in soil may increase, and the solute content in groundwater may change dramatically. The three media, with the coarse sand and medium sand diameter, are large, and the release coefficient is relatively large. When the same water level is rising and dropping, there is more water, and the migration is migrated. The nitrogen content is also greater. The increase of the water level will dissolve the adsorbed nitrogen in the aeration zone. The particle size of fine sand is the slightest, its ability to adsorb nitrogen is the strongest, and the nitrogen in fine sand is not easily transferred. In addition, the rise and fall of groundwater levels will change the redox

environment in the soil. When the water level rises, the aeration zone is filled with water, and the original oxidation environment becomes the reduction environment. When the water level drops, the aquifer releases moisture, and the initial reduction environment becomes the oxidation environment. Different soil environments are suitable for the survival of various microorganisms. The migration and transformation of nitrogen in soil media are closely related to microorganisms, adsorption-desorption, and solute transport. Nitrogen transformation is almost completed by a microbial-mediated redox reaction. In a large number of previous studies, the migration and transformation of three nitrogen by microorganisms mainly include nitrification, denitrification, anaerobic ammonia oxidation, dissimilatory reduction, and assimilation reduction.

Therefore, the activity of microorganisms is significant, and dissolved oxygen is an important index affecting its activity. The content of dissolved oxygen decreases and increases significantly with the rise and fall of the water level. After the water level rises, the aeration zone moves up, resulting in a decrease of the dissolved oxygen at the same position, the original aerobic environment becomes an anoxic environment, denitrification plays a leading role in this process, and the concentration of $NO_3^- - N$ decreases. When the water level drops, the capillary effect of the unsaturated zone weakens, the content of free water in soil pores decreases, the saturation decreases, and the contact area between pore water and air increases, resulting in the growth of the dissolved oxygen content, enhancement of oxidation, promotion of nitrification reaction, and rise of the soil $NO_3^- - N$ concentration. Among the three media, the particle size of coarse sand is the largest, the porosity of coarse sand is the largest, its nitrification and denitrification ability is the strongest, the content of free water is high, and $NO_3^- - N$ migrates downward quickly. Therefore, nitrogen changes more obviously with the fluctuation of the water level [32,33]. On the contrary, fine sand has the weakest nitrification and denitrification ability, a high bound water content, basically no downward migration of $NO_3^- - N$, and an inadequate response to groundwater level fluctuation.

The concentration of $NH_4^+ - N$ showed a downward trend as a whole. When the water level decreases, the dissolved oxygen content increases, the nitrification increases, the $NH_4^+ - N$ is transformed into $NO_3^- - N$, and the $NH_4^+ - N$ content in soil decreases. As the water level rises, denitrification is enhanced, $NO_3^- - N$ is transformed into $N_2O$ and $N_2$, and the total nitrogen in the soil is reduced [34]. Combined with the change process of $NH_4^+ - N$, when the water level increases, the concentration in coarse sand and fine sand medium shows an apparent upward trend. It is speculated that the hydrolysis injected into the soil column absorbed the $NH_4^+ - N$ adsorbed initially by soil particles, while the adsorption capacity of fine sand was vital, which was not conducive to the accumulation of $NH_4^+ - N$ in soil-free water. Hence, it changed little with the fluctuation of the water level. According to the change process diagram of $NO_3^- - N$ and $NH_4^+ - N$, in medium sand and fine sand media, the concentration of $NO_3^- - N$ and $NH_4^+ - N$ at the sampling point close to the surface layer is significantly higher than that at the more profound sampling point. With the decrease of the soil particle size, the vertical migration of nitrogen will also decrease.

## 5. Conclusions

Through this experiment, it is concluded that the impact of groundwater level fluctuations on the three media of coarse, medium, and fine sand is different. Groundwater fluctuations will affect various indicators in the soil, and then affect the migration and transformation of nitrate nitrogen, ammonium nitrogen, and nitrite nitrogen. In summary, the conclusions drawn are as follows:

- Dissolved oxygen in coarse, medium, and fine sand fluctuates obviously with the groundwater level and has an apparent negative correlation with the water level fluctuation. Coarse sand and medium sand obviously fluctuate with the water level;
- Groundwater level fluctuation can significantly affect the law of nitrogen migration and transformation in soil groundwater. When the water level drops, the $NO_3^- - N$

concentration increases, and $NH_4^+-N$ decreases. When the water level rises, the $NO_3^--N$ concentration decreases, and the $NH_4^+-N$ concentration increases. The variation of the $NO_3^--N$ concentration in the three media with the water level is medium sand > fine sand > coarse sand. The $NH_4^+-N$ concentration shows a downward trend, and the fluctuation of coarse sand and medium sand with the water level is more prominent;

- $NO_2^--N$ in the coarse sand and medium sand first increases with the fluctuation of the water level and then gradually reaches equilibrium. The sampling points below the water level in fine sand show a downward trend with the change of the water level, and then gradually reach equilibrium;

- Previous studies mainly focused on microbial nitrification and denitrification, and less on assimilation reduction, dissimilation reduction, and anaerobic ammonia oxidation. The research on interdisciplinary disciplines, such as hydrochemistry and molecular biology, is weak, and it is impossible to know precisely how microorganisms play a role in the migration and transformation of nitrogen in the fluctuation zone of the groundwater level. Therefore, in future research, hydrochemistry and molecular biology should be combined to find out what kind of reaction of nitrogen takes place through microorganisms in the fluctuation of the groundwater level, specifically nitrification, denitrification, dissimilatory reduction, or assimilative reduction.

Groundwater is an important part of water resources. Changes in hydrological conditions and groundwater exploitation activities will cause the rise and fall of groundwater levels. The fluctuation of the groundwater level will affect human production and life. The relationship between groundwater level fluctuations and humans will be closer in the future. Therefore, the study of groundwater fluctuation zone is of great significance.

**Author Contributions:** Conceptualization, Y.L. and J.Q.; methodology, Y.L., G.B. and X.Z; investigation, G.B. and X.Z.; writing—original draft preparation, Y.L. and G. B.; writing—review and editing, Y.L. and J.Q.; visualization, L.W. All authors have read and agreed to the published version of the manuscript.

**Funding:** This research was funded by the Key R & D and Promotion Projects in Henan Province (202102310012), the Doctoral Research Fund of North China University of Water Resources and Electric Power (40651), and the Key Project of Science and Technology Research of Henan Education Department(14A170006).

**Institutional Review Board Statement:** Not applicable.

**Informed Consent Statement:** Not applicable.

**Data Availability Statement:** Data was contained within the article.

**Conflicts of Interest:** The authors declare no conflict of interest.

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
