# Peer review of "Nitrogen Migration and Transformation Mechanism in the Groundwater Level Fluctuation Zone of Typical Medium"

_water, doi:10.3390/w13243626_

Round 1

Reviewer 1 Report

This manuscript focuses on the migration and transformation mechanism of nitrogen in groundwater level fluctuations, the research ideas in this paper are relatively clear, but the details of the experimental design need to be improved. Moreover, the analysis of the research results mainly focuses on the migration process of different nitrogen forms, and involves less transformation process. In addition, it is suggested that the author supplement and improve the microbial correlation analysis of nitrogen transformation in different forms and adsorption in transformation process. It is suggested to have a major revision before the acceptation for publication.

  1. It is suggested that the author unify the terms groundwater and coarse sand.
  2. Line 91, the section of 2.1. Experimental Materials, Nitrogen transformation is dependent on the action of microorganisms, the author used the original aquifer medium of the study site to carry out the experiment, whether the biological effect was considered? Is it sterilized?
  3. Line 91, the section of 2.1. Experimental Materials, Has the author performed a background analysis of nitrogen (NH4+, NO2-, NO3- and organic nitrogen) in the tested aquifer medium? Has the influence of leaching on nitrogen transformation been considered during the experiment?
  4. Line 120, the section of 2.3. Experimental design, Is light protection considered for the test soil column? Moreover, How is the fluctuation range of the water table determined? What is the actual groundwater depth of the sampling site? What is the fluctuation of water level in meters? It is suggested that the author explain it in detail.
  5. Line 120, the section of 2.3. Experimental design, What are the influent and effluent velocity during the whole experiment? Does the effluent velocity change when the groundwater level fluctuates? Many details about the experimental design need to be carefully supplemented by the author.
  6. Lin152, The section of 3 Results, how the author considers the quality control treatment for surface water and groundwater samples, the author should express for what were the quality control and quality assurance of the measurements? i.e. what were the blanks, standard reference materials, spikes, duplicates, triplicates, RSD, and/or error on measurements?
  7. According to Figure 4, the initial dissolved oxygen concentration of experimental influent water is 4mg/L-6mg/L, and the concentration of dissolved oxygen increases with the increase of sampling depth. What is the reason? experimental influent water is artificial configuration, the high concentration of dissolved oxygen is obtained by artificial aeration?
  8. In the analysis part of nitrogen concentration change, the author emphasizes that nitrogen transformation is the comprehensive result of medium adsorption and microbial action. However, there is no detailed explanation and analysis result to support the analysis of adsorption process and microorganism, so it is suggested that the author should supplement the relevant content of these two parts.
  9. Three types of media were used in the experiment, and the similarities and differences of nitrogen migration and transformation in the three media were suggested in the analysis of the results.

Reviewer 2 Report

The manuscript provides a valid research on nitrogen concentration and fluctiution in groundwater using an experimental and analogic approach. The authors need to specify better the application of their work in sub-surface hydrology and add relevant references. The manuscript is suitable for Water after implementation of all my spcific comments.

Specifc comments

Abstract

Line 11. “Analysed” better than “carried out” here.

Line 22. “Groundwater pollution”, would you like to mention which type of aquifers? Your research is applicable to certain types of aquifers

Introduction

Lines 26-28. Insert recent paper on notrogen concentration in both soil and groundwater due to fertilizers and lifestock

Medici, G., Baják, P., West, L. J., Chapman, P. J., & Banwart, S. A. (2021). DOC and nitrate fluxes from farmland; impact on a dolostone aquifer KCZ. Journal of Hydrology595, 125658

Lines 45-46. “In the 1990s…..fluctuation”, add papers related to this topic and time span

Lines 25-89. You mention that your research is also applicable to aquiefrs not only in soil. Would you like to explian in detail that your research finds application in un-conslidated siliciclastic deposits of fluvial origin? I’m talking about the typical aquifers of quaternary age above the bedrock

Lines 88-89. State aim and the specific objectives of your research.

Material and methods

Lines 146-151. More detail on the pH meter (bids, manifacturer etc etc) and experimental procidures. You need to expand the sub-paragraph 2.4

Results

Lines 302-306. Insert reference to figures when you describe rise and drops of concentrations and values

Line 328. You say “nitrification” twice, fix the issue

Discussion

Line 368. What do you mean by “soil environments”? Maybe, soil types? Explain with more detail

Lines 377-385. Split these parts in multiple-sentences

Conclutions

Lines 408-432. Add two sentences at the biginning of the paragraph and a conclusive remark after your bulletin points

References

Line 444. Insert the references suggested above

Figures and tables

Increase graphic resolution of Figures 1 and 3

Round 2

Reviewer 1 Report

(1)In this paper, functional genes of nitrification and denitrification were analyzed. However, we did not see any detection results of functional genes, and what is the spatial distribution of functional genes in the test soil column? What is the process of influence? It is suggested that the authors supplement relevant data and analysis.

(2)As for the comments of Point 4, I want to know the actual depth and fluctuation range of groundwater level in the sampling area, but not in the test design.

(3)As for Point 7, the author does not explain the reason why the concentration of dissolved oxygen increases with the increase of depth, but draws the conclusion that the dissolved oxygen can be ignored.
